

**Frequency and distribution of winter melt events from passive microwave satellite data in**
**the pan-Arctic, 1988-2013**
Libo Wang[1], Peter Toose[1], Ross Brown[2], and Chris Derksen[1]
[1] Climate Processes Section, Climate Research Division, Environment and Climate Change Canada,
Toronto, Ontario, Canada
[2] Climate Processes Section, Climate Research Division, Environment and Climate Change
Canada@Ouranos, Montreal, Québec, Canada
Correspondence to: Libo Wang (libo.wang@canada.ca)
**Abstract**
This study presents an algorithm for detecting winter melt events in seasonal snow cover based
on temporal variations in the brightness temperature difference between 19 and 37 GHz from
satellite passive microwave measurements. An advantage of the passive microwave approach is
that it is based on the physical presence of liquid water in the snowpack, which may not be the
case with melt events inferred from surface air temperature data. The algorithm is validated
using in situ observations from weather stations, snowpit surveys, and a surface-based passive
microwave radiometer. The results of running the algorithm over the pan-Arctic region (north of
50º N) for the 1988-2013 period show that winter melt days are relatively rare averaging less
than 7 melt days per winter over most areas, with higher numbers of melt days (around two
weeks per winter) occurring in more temperate regions of the Arctic (e.g. central Quebec and
Labrador, southern Alaska, and Scandinavia). The observed spatial pattern was similar to winter
melt events inferred with surface air temperatures from ERA-interim and MERRA reanalysis



datasets. There was little evidence of trends in winter melt frequency except decreases over
northern Europe attributed to a shortening of the duration of the winter period. The frequency of
winter melt events is shown to be strongly correlated to the duration of winter period. This must
be taken into account when analyzing trends to avoid generating false increasing trends from
shifts in the timing of the snow cover season.
**1. Introduction**
Snow cover is important in Arctic climate and ecological systems and has decreased in areal
extent and duration especially during the spring period in response to rapid Arctic warming in
recent decades [Brown and Robinson, 2011; Derksen and Brown, 2012; Callaghan et al. 2012].
The conventional wisdom is that Arctic warming will result in an increase in the frequency and
duration of winter melt events which may also include rain-on-snow (ROS) events. These winter
melt/refreeze events modify the physical properties of snow (albedo, density, grain size, thermal
conductivity), generate winter runoff [Bulygina et al., 2010; Johansson et al., 2011] and can
result in potentially significant impacts on the surface energy budget, hydrology and soil thermal
regime [Boon et al., 2003; Hay and McCabe, 2010; Rennert et al., 2009]. The refreezing of melt
water can also create ice layers that adversely impact the ability of ungulate travel and foraging
[Hansen et al., 2011; Grenfell and Putkonen, 2008], and exert uncertainties in snow mass
retrieval from passive microwave satellite data [Derksen et al., 2014; Rees et al., 2010]. Winter
warming and melt events may also damage shrub species and tree roots, affecting plant
phenology and reproduction in the Arctic [Bokhorst et al., 2009; AMAP, 2011]. However, little
is known about the spatial and temporal variability of winter melt events at the pan-Arctic scale.




Winter melt events are rare extreme events over most of the Arctic and are sporadic in time and
space [Pedersen et al., 2015]. These events are linked to intrusion of warm air from southerly or
southwesterly flow, may be associated with rain and/or freezing rain, and typically last for
several days.  Previous studies [Cohen et al. 2015; Rennert et al 2009] have shown that the
synoptic conditions associated with these events project strongly onto larger modes of
atmospheric circulation.

Microwave remote sensing measurements are very sensitive to the presence of liquid water in
snow. Dry snow is a mixture of air and ice. Because the permittivity of water is much higher than
those of air and ice at microwave frequencies, the introduction of even a small amount of liquid
water (0.5%) in snow can increase the permittivity of snow by over an order of magnitude
[Ulaby et al., 1986]. This increases absorption and reduces the penetration depth, which in turn
results in a large increase in brightness temperature ($T_B$) and decrease in radar backscatter.
Satellite active and passive microwave measurements have been widely used for snow melt
detection over various components of the Arctic cryosphere during the spring melt period
[Markus et al., 2009; Tedesco, 2007; Kim et al., 2011, Wang et al., 2011]. Only a few satellite
studies have focused on winter melt detection, and are mainly based on active microwave
satellite data [Bartsch et al., 2010; Wilson et al., 2012; Semmens et al., 2013] for specific regions
and limited time periods. Here we develop an algorithm to detect winter melt from satellite
passive microwave (PMW) data over pan-Arctic land areas north of 50º N for the period 1988-

66    2013.




Winter melt and ROS events can also be inferred from surface weather observations [Groisman
et al., 2003; McBean et al., 2005; Pedersen et al., 2015], reanalyses [Cohen et al. 2015; Rennert
et al., 2009], or reanalysis-driven snowpack models [Liston and Hiemstra, 2010]. In most of
these studies, winter melt events are assumed to occur when the daily air temperature exceeds a
certain threshold. For example, Groisman et al. [2003] defined a thaw day as a day with snow on
the ground when the daily mean air temperature is above -2º C. Inferring thaw events from
surface air temperatures in this way  does not consider the energy balance of the snowpack.  In
addition, reanalysis datasets can contain important biases and inhomogeneity over the Arctic [e.g.
Rapaic et al. 2015] that will impact the spatial and temporal frequency of the inferred winter
thaw events. The advantage of the passive microwave approach described above is that melt
events are directly linked to the appearance of liquid water in snow which drives changes in
snowpack properties relevant to Arctic ecosystems. The brightness temperature series is also
considered to be consistent over the 1988-2013 period as it is derived from near identical
spaceborne sensors.

Previous studies have linked field observations of ice layer formation from ROS events with
satellite measurements [Bartsch et al., 2010; Grenfell and Putkonen, 2008], but few studies have
showed links between satellite measurements and in situ observations of changes in snow
properties from melt/refreeze events [Nghiem et al., 2014]. The passive microwave satellite data
used to detect changes in snow properties due to ROS or melt/refreeze events are in coarse
resolutions (10-25 km) with twice daily overpasses at the high latitudes. Thus melt events of
short duration or limited spatial distribution may not be detectable. The objectives of this study
are to (1) develop an algorithm for winter melt detection from PMW data, and (2) to characterize



winter melt events detectable by PMW at the satellite scale using surface–based PMW
radiometer measurements and snowpit surveys collected during field campaigns. These results
are compared to winter melt detection results inferred from near surface air temperature fields
from two commonly used reanalysis datasets. Trends in PMW-derived winter melt frequency
over the period 1988-2013 are presented along with a demonstration of the impact on trend
results of using a fixed winter period for defining the snow season.

**2. Data and Methods**

**2.1. Satellite passive microwave data**

This study uses $T_B$ data from the Special Sensor Microwave/Imager (SSM/I, 1987–2008), and
the Special Sensor Microwave Imager/Sounder (SSMIS, 2009 to present) re-projected to 25 km
EASE-Grid available from the National Snow and Ice Data Center in Boulder, Colorado
[Armstrong et al., 1994]. These sensors provide a continuous time series of $T_B$ since 1987 (Table
1). We do not perform sensor cross calibration given that only small differences were found
between sensors [Abdalati et al., 1995; Stroeve et al., 1998; Cavalieri et al., 2012]. Since our
melt detection algorithm (described below) only uses the relative change in the temporal
variations in $T_B$, slight offsets in absolute $T_B$ between sensors should not affect algorithm
performance. The gaps in the data are filled by linear interpolation from adjacent days. Vertically
polarized $T_B$ from afternoon overpasses are utilized to increase the likelihood of observing melt
events, rather than morning overpasses. Due to large temporal gaps in the early SSM/I record
(pre-1987), the time series used begin in the fall of 1988 and extend to 2014 (Table 1). Although



horizontal polarized measurements are more sensitive to ice lenses within the snowpack
[Derksen et al., 2009; Rees et al., 2010], there is not much difference between the two
polarizations for melt detection and we use vertically polarized measurements to be consistent
with Wang et al. [2013].

**2.2. Winter melt detection method for PMW**

As the purpose of this study is to detect winter melt events, the winter period duration (WPD) is
defined as occurring between the main snow onset date (MSOD) in the fall (beginning of
continuous dry snow cover on the ground) and the main melt onset date (MMOD) in the spring
(beginning of frequent melt/freeze diurnal cycles). Figure 1 illustrates the steps involved in
detecting melt events for the WPD, based on the temporal variations in the difference of the
brightness temperature ($T_BD$) between 19 and 37 GHz and a 37GHz $T_B$ threshold. For dry snow
conditions, as snow accumulates $T_BD$ increases due to the larger scattering effect of the
microwave signal by snow grains at 37 GHz versus 19 GHz [Chang et al., 1987]. Upon the
appearance of liquid water in snow, $T_B$ increases at both frequencies and results in a drop in $T_BD$,
to similar magnitudes seen in snow free conditions, but will quickly revert back to dry snow $T_BD$
levels once the snow re-freezes allowing for the detection of melt/re-freeze events (Figure 2).

The purpose of determining MSOD is to capture the earliest start date of the continuous dry
snowpack. The MSOD is determined as the first date when (1) $T_BD \geq Tsn$ (a threshold = mean
July $T_BD$ + 3.5 K) for 7 out of 10 days and (2) $T_B37v < 253$ K for 10 out of 11 days (Figure 1).
The thresholds and conditions were optimized by comparing the PMW determined MSOD to





daily snow depth observations from the Global Surface Summary of the Day dataset archived at
the National Climate Data Center (http://www.ncdc.noaa.gov). The $T_B$ criterion in (2) is applied
to exclude periods with $T_B D$ fluctuations related to early season freeze/thaw cycles because of its
sensitivity to the presence of liquid water in the snow (see below for its derivation).

MMOD is determined following Wang et al. [2013]. Their algorithm was based on temporal
variations in $T_B D$ relative to the previous 3-day average $T_B D$ (referred as M hereafter). Melt
onset was detected if the difference in M and daily $T_B D$ was greater than a threshold ($TH_{old}=$
0.35*M) for four or more consecutive days. Based on trial and error, the MMOD detection
algorithm in Wang et al [2013] is modified here to detect mid-winter melt events that are
typically of shorter duration. Firstly, the threshold is modified slightly from $TH_{old} = 0.35*M$ to
$TH_{new} = 0.4* M$ since the goal is to detect melt events with one or more days of duration (instead
of four days as in the previous study, thus a more strict threshold here), and secondly, a $T_B 37v$
threshold condition is added following Semmens et al. [2013] to mitigate false detection due to
$T_B D$ changes not related to melt (e.g. noise). The resulting expression for winter melt event
conditions is $(M-T_B D) > TH_{new}$ and $T_B 37v \geq 253$ K for one day (Figure 1, referred as the winter
$T_B D$ algorithm hereafter). The $T_B 37v \geq 253$ K condition was obtained by evaluating a range of
$T_B 37v$ values from 250-255 K, at 1 K increments to identify the threshold most sensitive to the
presence/absence of liquid water in snow. This was inferred from  histograms of daily maximum
(Tmax), mean (Tm), and minimum (Tmin) air temperatures for days detected as melting at all
available stations during 2000-2007 (see locations in Figure 5b, ~5100 observations in total).
The results show that for $T_B 37v = 253$ K, Tmax is $\geq 0°$ C for nearly 96% of cases, Tmin is $< 0°$
C for 94%, and Tm is $\geq 0°$ C for 80%. This suggests that the PMW-detected winter melt events





are consistent with diurnal positive air temperature events, while most of the events (80%)
probably last multiple hours thus corresponding to days with Tm $\geq$ 0º C. If a melt event is
detected within 10 days from the MMOD, then it is not considered a mid-winter melt event, but
rather a preliminary melt event to the MMOD and is excluded from the analysis.

An example of the performance of the winter $T_BD$ algorithm is shown in Figure 2 for a case at
Pudasjarvi, Finland (65.4º N, 26.97º E) during the 2013- 2014 winter. At Pudasjarvi station, the
snow depth first became greater than 0 cm on day of year (DOY) 291 of 2013. The snow depth
was mostly less than 10 cm for days 291 to 332, with two periods of no snow on the ground
while Tmax fluctuated around 0º C. The PMW detected MSOD was on DOY332, corresponding
within 1 week of the date of continuous snow cover above 10 cm observed at the station (Figure
2b). MMOD was detected on DOY64 of 2014, however, there was still snow on the ground until
DOY108, typical of high latitude snow cover where melt onset is followed by the spring thaw,
which is a sustained period with high diurnal air temperature variation where the snowpack is
melting during the day and refreezing at night. At the end of this melt–refreeze period, the
snowpack may be actively melting both day and night until snow disappearance which can take
several weeks [Semmens et al., 2013]. During winter 2013-2014, 20 melt days in total were
detected at Pudasjarvi, all corresponding to days with Tmax $\geq$ 0$^{o}$ C. However, not all days with
Tmax $\geq$ 0$^{o}$ C are detected by PMW as melting, for example DOY351-352, for reasons which
will be explained further in the validation section.

The winter $T_BD$ algorithm is applied to time series of $T_B$ for each winter period over the period
1988-2013. The WPD varies at each pixel and is determined by MSOD and MMOD as described



above. This approach is referred to as "PMW-varying" in the following analysis. Since we focus
on melt events during the winter period, the $T_BD$ algorithm is only applied to pixels with MSOD
detected before the end of December and with MMOD later than March 1st, i.e. with WPD > 60
days. The PMW-varying approach is internally consistent in that it takes account of annual
variations in winter temperature and snow cover. This is not the case for analysis using a fixed
"winter" window where spurious trends can be created from changing seasonality (i.e. earlier
snow melt). To highlight this, a fixed window approach is also applied ("PMW-fixed") where the
$T_BD$ algorithm is applied to time series of $T_B$ from November to April. The results presented in
the following sections are from the PMW-varying method unless explicitly indicated otherwise.
Since the microwave response of melt on permanent snow and ice is different from seasonal
terrestrial snow cover, we mask out the Greenland Ice sheet and glaciers in our analyses.

**2.3. Winter melt detection for reanalysis datasets**

Winter melt event information from the 0.75° x 0.75° degree latitude/longitude ERA-interim
(ERA-I) [Dee et al., 2011] and the 1/2° latitude by 2/3° longitude MERRA [Rienecker et al 2011]
reanalyses were used to evaluate the melt event climatology generated by the PMW method.
Melt events in the reanalyses are inferred from 6-hourly air temperatures over the same period as
the satellite data. For the comparison, a winter thaw event is defined as a period of above-
freezing daily mean air temperature occurring during the period dominated by below-freezing air
temperatures (defined by 0° C crossing dates obtained with a centered 30-day moving average of
daily mean air temperature).  This is analogous to the "PMW-varying" method described above.
An additional condition is imposed of at least 10 cm snow depth for ERA-I or 4 mm SWE for



MERRA on the ground to obtain results comparable to the PMW method of detection over snow
covered ground. The mean daily air temperature is the average of the 00, 06, 12 and 18 UTC
values.  Snow depths for ERA-I are taken from the  daily snow depth reconstruction described in
Brown and Derksen [2013] to avoid various inconsistencies with the snow depths in the
reanalysis.

**2.4. In situ field observations and methods**

The satellite-based winter melt detection algorithm is validated with surface–based PMW
radiometer measurements along with near surface air/snow temperature observations recorded on
April 12$^{th}$-13$^{th}$, 2010 during a field campaign near Churchill, Manitoba, Canada [Derksen et al.,
2012]. A modified version of the winter $T_BD$ algorithm is applied to the surface-based
radiometer measurements due to the continuous nature of the data. We simply used the average
$T_B$ values from the stable pre-melt period as our reference frozen $T_BD$ value instead of previous
3-day average.

Furthermore, we try to characterize winter melt events detectable by the winter $T_BD$ algorithm
using snowpit surveys recorded during multiple PMW snow measurement campaigns conducted
between 2005 and 2010 in both the boreal forest and tundra environments of Canada (Table2).
The number of satellite detected melt events for the specific EASE-Grid pixels surrounding the
snow pit locations are compared to the number of melt forms/ice formations identified within the
snowpack. A melt feature identified lower (closer to the ground) is consider an early winter event,
while those melt features identified closer to the surface of the snow are considered more recent



events. An example of the coincident satellite, air temperature and snow pit information for a
survey site near Thompson, Manitoba is shown in Figure 3. Hourly air temperatures from
weather stations in the vicinity of the snow pits (within 70 km), are examined to identify if and
when a melt event occurred in the region, how long the melt event lasted, what the average
temperature was for the duration of the event and what the minimum, maximum and average 36
hour air temperatures were preceding the melt event. Results of the field evaluation are presented
in Section 3.1

**3. Results**

**3.1. Field evaluation of melt algorithm**

Figure 4 illustrates the time series of the surface-based radiometer $T_B$ and temperature
measurements recorded during the April 12th-13th Churchill melt event. The area in green
highlights the period for which the modified $T_B D$ algorithm identified the melt event. As the near
surface air temperatures approached 0º C, $T_B$ increased rapidly at both the 19 and 37 GHz. The
detected melt onset occurred ~ 40 minutes after the 11 cm and 7 cm air temperatures crossed the
0° C threshold and 25 minutes before the 2 m air temperature exceeded 0° C, likely due to
radiant heating from the sun to the snow surface and the boundary layer air temperature probe.
The -1 cm snow temperature didn't reach 0° C until three hours after the detected melt onset. The
influence of radiant heating becomes obvious during the late afternoon/early evening, upon
sunset (~1900 hr local) the snowpack and boundary layer air temperatures all drop below 0° C,
closely followed by a gradual drop in the $T_B$ signal even while the 2 m air temperatures are still



positive. Compared to the rapid increase in $T_B$ during the melt onset, the more gradual decrease
in $T_B$ is likely due to the mixed effects of uneven re-freezing of the snow surface and delayed
freezing of sub-surface liquid water.

The validation results using snowpit data are summarized in Table 2. The performance of the
winter $T_B$D algorithm is highlighted in green for a successful melt detection and in red for a
failed detection. The results suggest that a successful detection is likely when the melt duration
last for multiple hours (>6 hours) or multiple days, and/or the melt event has been preceded by
warm air temperatures that have warmed the snowpack to near melting conditions (previous
day's Tmax > -3° C). In these situations, it is common for melt features to form within the
snowpack. The algorithm does not reliably identify short duration melt events or events that
occur immediately after extremely cold air/snowpack temperatures (previous 36 hour minimum
air temperature < -13° C). In these instances, the snowpack likely has enough thermal inertia to
remain within a frozen state for the whole duration of the melt event, or very quickly return to a
frozen state and thus liquid water is not detectable with satellite $T_B$.

The winter $T_B$D algorithm is also not well suited to detect ROS events and the subsequent
development of ice layers within the snowpack. The Daring Lake [Rees et al., 2010] and
LaGrande IV melt events presented in Table 2 were coincident with ROS, but were both quickly
followed by cold air temperatures leading to the re-freezing of the liquid water and were thus not
detected. The winter $T_B$D algorithm is very sensitive to liquid water within the snow, but does
not necessarily capture all events that can create melt features within the snowpack, largely due
to the timing of the satellite overpass (~1800 h local) and the coarse resolution (25 km).




**3.2. The spatial distribution of winter melt events**


Figure 5 shows the PMW-derived MSOD, MMOD, and WPD during the 1988-2013 period. On

average, continuous snow cover starts in the Canadian Arctic islands and high elevation regions

of the Arctic in September and progresses to the open tundra in October (Figure 5a). By

November, most of the areas north of 50º N are covered by snow except for some temperate

maritime and lower latitude regions where continuous snow cover sets in December. The spring

main melt onset starts at lower latitudes in March, progresses to the boreal forests and tundra in

April/May, and reaches the high Arctic in June (Figure 5b), giving rise to spatial variability in

the duration of the winter period from one to seven months on average (Figure 5c A pixel-wise

definition of winter period for winter melt detection is required to account for this spatial

variability as well as the temporal variability from year-to-year fluctuations in snow cover.

288

During the 26 winters, melt occurred at least once everywhere north of 50º N using the PMW-

varying window method (Figure 6a). However, the average number of melt days is less than one

week per winter for most areas, with more melt days (around two weeks per winter) occurring in

areas with a relatively long snow season and more temperate winter climates (e.g. central Quebec

and Labrador, southern Alaska, and Scandinavia). The spatial distribution patterns of NMD from

ERA-I (Figure 6c) and MERRA (Figure 6d) generally agree with that from PMW. However,

ERA-I detects about one week more melt days on average in most areas , while MERRA detects

less melt days in Quebec and central Canada relative to  PMW. Both ERA-I and MERRA detect

more melt days in southern Alaska and western North America (NA). These are relatively deep



snowpack regions where melt may not occur in short periods of freezing air temperatures due to
the thermal inertia of the snowpack. Compared to the PMW-varying window method (Figure 6a),
there are many more melt days detected using the PMW-fixed window method (Figure 6b),
especially in the relatively temperate climate regions (e.g. northern Europe and lower latitudes of
NA and Russia) where the WPD is relatively short and thus limits the possible number of melt
days to be detected.

Figure 7 shows the monthly mean NMD from October to May during the period 1988-2013.
Winter melt events mainly occur in the fall (October-November) and spring (April-May) months
at high latitudes (>60º N) where continuous snow starts early and melts late in some years
(Figure 5). During November to March for the period 1988-2013, no winter melt events are
detected across large areas of Siberia and the Canadian and the Alaskan tundra where the
monthly surface air temperature (SAT, from the Climatic Research Unit (University of East
Anglia) CRUTem4 dataset [Jones et al., 2012]) is usually lower than -20º C (Figure 8). On
average, April has the maximum extent and duration of winter melt events (Figure 7).

**3.3. Changes in snow cover and winter melt events**

The Mann-Kendall method is used for trend analysis taking into account serial correlation
following Zhang et al. [2000]. Trends are only computed at grid cells with melt events detected
in at least 12 winters. The PMW-derived estimates of changes in snow cover (MSOD, MMOD,
and WPD) over the 1983-2013 period are shown in Figure 9. Most of the Arctic exhibits later
snow onset trends, particularly over Scandinavia, western Russia, Alaska, Quebec and most



coastal areas (Figure 9a). The timing of the spring main melt onset date tends to be earlier over
most of the Arctic except for northern Europe and western NA (Figure 9b). As a result, there are
significant decreasing trends of more than 30 days in the duration of winter period over most of
the Arctic (Figure 9c).

Over the study period, there are few significant trends in NMD over the Arctic (Figure 10a), and
where there are significant trends, these tend to be dominated by decreases over northern Europe.
The spatial distribution patterns of NMD trends contrast markedly between the PMW-varying
and the PMW-fixed results (Figure 10a and b). Trends from PMW-fixed are dominated by
increasing trends in NMD over most of the Arctic except for northern Europe. Trends from the
reanalyses are not shown because the annual winter thaw frequency series from ERA-I and
MERRA are not always consistent over the 1988-2013 period in some regions. For example over
northern Quebec (not shown) the two series are well correlated over the period from 1980-2001
(r=0.75) but diverge markedly after 2001 when numerous changes in data assimilation streams
occurred in both reanalysis datasets [Rapaic et al. 2015]. This underscores the advantage of the
PMW melt detection approach where a consistent time series of $T_B$ are obtained from near
identical sensors.

**4. Discussion and conclusions**

An algorithm for detecting winter melt events using satellite PMW measurements is developed
and evaluated using in situ observations at weather stations and field surveys. The use of the high
resolution (both spatially and temporally) surface-based radiometers and temperature profile data



highlight the fact that passive microwave radiometers are particularly sensitive to minute
amounts of liquid water present at the snow surface as is evident by the dramatic change in the
radiometric signal observed even when the recorded snow temperature (at 1 cm below the
surface) are still below 0° C. The winter $T_BD$ algorithm has a higher success rate when the melt
duration last for multiple hours/days and/or the melt event has been preceded by warm air
temperatures. The algorithm does not reliably identify short duration melt events, ROS, or events
that occur immediately after or during extremely cold air/snowpack temperatures.

During the period 1988-2013, winter melt occurred at least once everywhere for north of 50ºN.
On average, melt occurs less than one week per winter for most Arctic areas, with more melt
days (approximately two weeks per winter) occurring in areas with relatively long snow season
and temperate climate. Winter melt events are not detected in some areas of Siberia and the
Canadian and the Alaskan tundra where the monthly SAT is usually lower than -20ºC. The
spatial distribution patterns of NMD are in general consistent from the reanalysis datasets (ERA-
I and MERRA) and PMW, while the detected NMDs are different probably due to biases in the
reanalysis datasets and the different methodology used to infer melt events.

Over the period 1988-2013, most of the Arctic exhibits later snow onset in fall, earlier melt onset
in spring, and thus decreasing duration of winter period. There are no significant trends in NMD
over most of the Arctic except for norther Europe where there are decreasing trends. The number
of melt days was observed to be significantly correlated with the duration of winter period over
most of the Arctic, particularly in regions where interannual variability in snow cover is higher
(Figure 11). Thus observed significant decreasing trends in WPD are playing a role in the



observed lack of significant increasing trends in NMD. A similar conclusion was reached by
Cohen et al. [2015] in a study analyzing ROS event trends from reanalyses. They also found that
the frequency of ROS events was correlated to large-scale modes of atmospheric circulation
which contributes to regional-scale variability in ROS trends. Another contributing factor to the
lack of increasing winter melt trends is the seasonal pattern of warming over Arctic land areas
during 1988-2013, which is dominated by warming in the snow cover onset period (September -
November) with comparatively little warming in the winter (December - February) and spring
(March - May) period (Figure 12).

There is field evidence of changes in snowpack density and ice layers from a number of locations
in the Arctic that is supported by an increased frequency of winter thaw events [Chen et al., 2013;
Groisman et al., 2003; McBean et al., 2005; Johansson et al., 2011]. However, winter thaw
events in some of these studies were inferred from air temperature observations [Groisman et al.,
2003; McBean et al., 2005], which is different from results detected by PMW measurements.
The lack of significant increasing trends in winter melt events observed in this study is also
likely related to the relatively short period of data available for analysis and the dynamic
mechanisms generating winter thaw and ROS  events which tend to produce more random and
chaotic environmental responses [Trenberth et al. 2015; Cohen et al. 2015]. This is underscored
by trend analysis of annual numbers of winter thaws events in ERA-I and MERRA over a longer
1980-2014 period (not shown) which revealed that locally significant increasing trends were only
observed at 1% of snow covered land points in MERRA and 2% in ERA-I.





As previously pointed out in Figure 10b, the frequency of winter melt events is strongly
influenced by the method used to define WPD.  A spatially and temporally varying definition of
WPD is required as the use of a fixed window generates artificial NMD trends from changes in
the timing of the snow cover season. This is further demonstrated in Figure 13 where monthly
NMD trends are computed using a fixed WPD of November-April. The results clearly
demonstrate that increases in NMD are being driven by trends during the snow cover shoulder
seasons of November-December and March-April and not the main winter period. A number of
studies reporting increasing NMD trends used fixed winter periods in their analysis [e.g.
Groisman et al., 2003; McBean et al., 2005].

The major advantage of the PMW winter melt event method presented here is that it is based on
physical processes in the snowpack (melt/freeze), unlike thaw events inferred from air
temperature observations that may or may not be associated with snowpack melt processes
depending on the thermal inertia of the snowpack. The PMW series is also consistent over time
unlike some reanalysis datasets. There is little evidence of significant trends in winter melt
frequency during the 1988-2013 period over most of the Arctic except for northern Europe
(decreasing trends).


**Acknowledgements.** The In-situ snow survey data used in this study was the result of multiple
campaigns over many years supported by numerous organizations which have provided direct
funding, logistical support or have contributed with people in the field. There have been too
many individual contributions to list them all here, so instead we will thank their affiliations



and/or their funding sources which include: Environment and Climate Change Canada, the
Canadian Space Agency, University of Waterloo, Université de Sherbrooke, Wilfrid Laurier
University, the Churchill Northern Study Centre, the Aurora Research Institute, the Canadian
Foundation for Climate and Atmospheric Science, Manitoba Hydro, the Northwest Territories
Power Corporation and Indian and Northern Affairs Canada. The following data centers are
acknowledged for providing data: The National Snow and Ice Data Center for passive
microwave satellite data, the National Climate Data Center for the Global Summary of the Day
dataset, the Climatic Research Unit - University of East Anglia for the CRUtem4v gridded air
temperature data, the European Centre for Medium-Range Weather Forecasts (ECMWF) for the
ERA-interim data, and the Global Modeling and Assimilation Office (GMAO) at NASA
Goddard Space Flight Center for MERRA data. The authors would like to thank Anne Walker
for providing helpful comments to an early version of the manuscript.

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



**Table 1.** Data periods for the different satellite passive microwave radiometers and overpass used for melt detection in this study.

| Satellite | Start Date | End Date | Overpass |
|-----------|-----------|----------|-----------|
| F-08 SSM/I | Jul 1988 | Dec 1991 | Descending |
| F-11 SSM/I | Jan 1992 | May 1995 | Ascending |
| F-13 SSM/I | May 1995 | Dec 2008 | Ascending |
| F-17 SSMIS | Jan 2009 | present | Ascending |



**Table 2.** Performance summary of the satellite melt detection using the winter $T_BD$ algorithm at snowpit survey sites across Canada, characterized with coincident nearby weather station air temperatures.

| Survey Site | | Snowpit Feature Depths (cm) | | | | Satellite Melt Detection | | Weather Station Air Temperature (°C) | | | | | |
|---|---|---|---|---|---|---|---|---|---|---|---|---|---|
| Weather Station / Year of Survey | Lat/Lon | Pit Depth | Melt Feature Height Above Ground | | DOY | DOY | Reason for Success/Failure | DOY | Melt Event # of HRS | Avg. Temp | Previous 36 HR Avg. Temp | Min. Temp | Prior Day Max. Temp |
| Thompson, MB 2005 | 56.016N 97.260W | 53 | Melt-freeze crust | 9-8 | 070 | 321 | Warm snow | 321 | 27 | 0.37 | 1.35 | -4.5 | 6.8 |
| | | | | 45-43 | | 034 | Warm snow | 033 | 8 | 1.44 | -2.69 | -5.5 | -1.8 |
| Gillam, MB 2005 | 57.020N 94.140W | 63 | Melt-freeze crust | 53-52.5 | 070 | 034 | Warm snow | 033 | 9 | 0.49 | -5.75 | -10.5 | -2.7 |
| Rae Lakes, NT 2006 | 63.882N 115.072W | 72 | Ice layer | 36 | 094 | Not Detected | Cold snow | 082 | 10 | 3.7 | -7.3 | -17.9 | 6.5 |
| | | | Melt-freeze crust | 62 | | | | | | | | | |
| | | | Sun crust | 72 | | | | 092 | 1 | 1.1 | -11.03 | -28.3 | -1.9 |
| Daring Lake, NT 2007 | 64.867N 111.573W | 48 | Ice layer | 48-47.5 | 100 | Not Detected | Rain event / Cold snow | 098 | 2 | 0.3 | -6.47 | -13.62 | -6.4 |
| Fort McPherson, NT 2008 | 67.569N 133.618W | 54 | No Lower Layer Melt Features Present | | 097 | Not Detected | Cold snow | 020 | 26* | 3.7* | -25.0* | -27.1* | -23.6* |
| | | | Ice layer | 41 | | 093 | Warm Snow | 093 | 32* | 2.9* | -3.57* | -13.0* | 6.1* |
| | | | | 49 | | | | | | | | | |
| | | | Melt-freeze crust | 54-53.5 | | 096 | Warm Snow | 095 | 4* | 2.88* | -0.83* | -7.5* | 4.7* |
| LaGrande IV, QC 2009 | 53.648N 73.875W | 72 | Melt-freeze crust | 39.5-39 | 078 | Not Detected | Cold snow | 362 | 5 | -0.3 | -11.20 | -27.7 | -6.3 |
| | | | Ice layer | 70-69.5 | | Not Detected | Rain event / Cold Snow | 076 | 17 | 2.45 | -19.60 | -33.4 | -10.6 |
| Churchill, MB 2010 | 58.7364N 93.8227W | 69 | Ice layers - multiple | 54-45 | 102 | 090 | Warm snow | 090 | 6** | 0.5** | -2.83** | -5.1** | -1.92** |
| | | | Melt-freeze/rain crust | 69-66 | | 099 | Warm snow | 099 | 13** | 5.4** | -1.32** | -9.31** | 8.76** |

\* Indicates that the weather station data is available only during daylight hours (recorded by observer), thus average values are not comparable to other stations
\*\* Indicates that air temperatures from a local meteorological station were used instead of the Churchill Climate Station (local met station was closer to the snowpit)





**Figure 1.** Schematic flow chart of the winter $T_BD$ melt detection method for PMW satellite data.



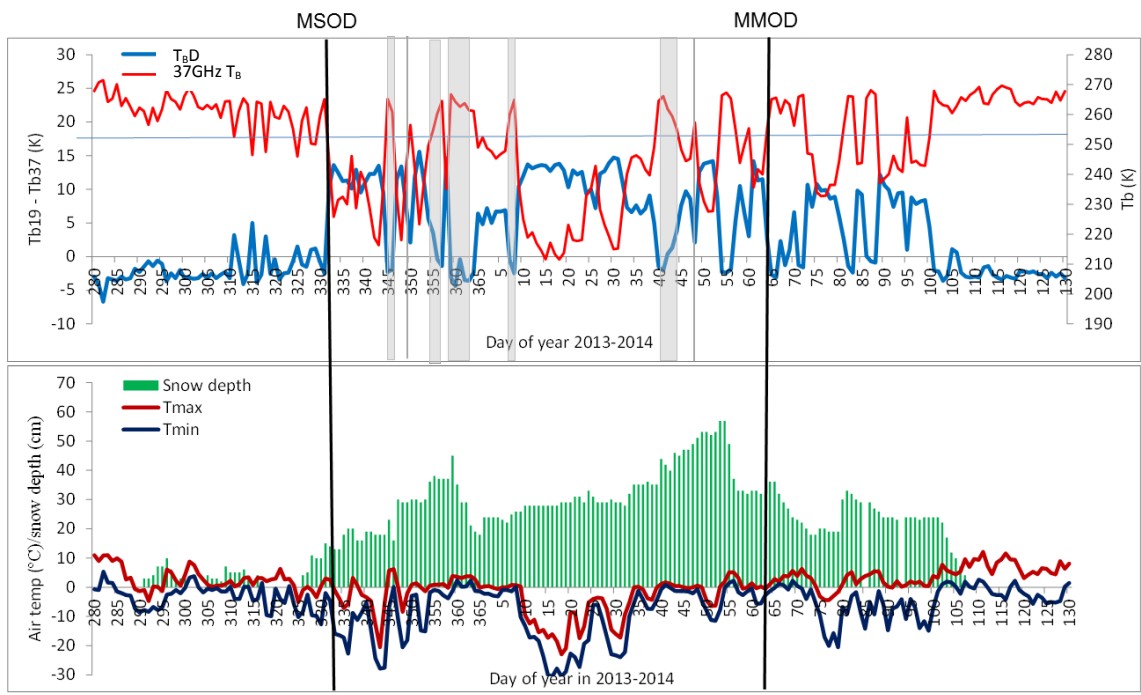

**Figure 2**. Example of time series of SSM/I $T_B$D (a) and daily surface air temperature (ºC) /snow depth (cm) (b) at Pudasjarvi, Finland (65.4ºN, 26.97ºE) during the 2013- 2014 winter. The vertical grey lines/bars in (a) represent melt events detected by satellite.



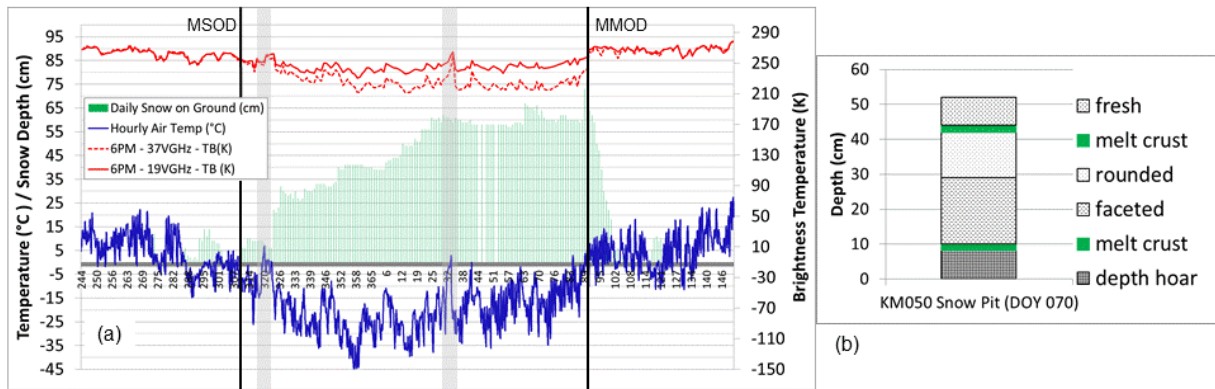

**Figure 3.** (a) Time series of hourly air temperature and daily snow depth and $T_B$ at the Thompson, Manitoba Meteorological Station from Sep. 2004 to May 2005; the shaded grey bars highlight the timing of the melt events detected by the PMW satellite data. (b) Snow stratigraphy from the KM050 snow pit site surveyed on DOY097. Note that both the early season and recent melt crusts observed in the snowpit agree reasonably well with the timing of two winter melt events recorded at the Thompson airport and detected by the PMW satellite data.





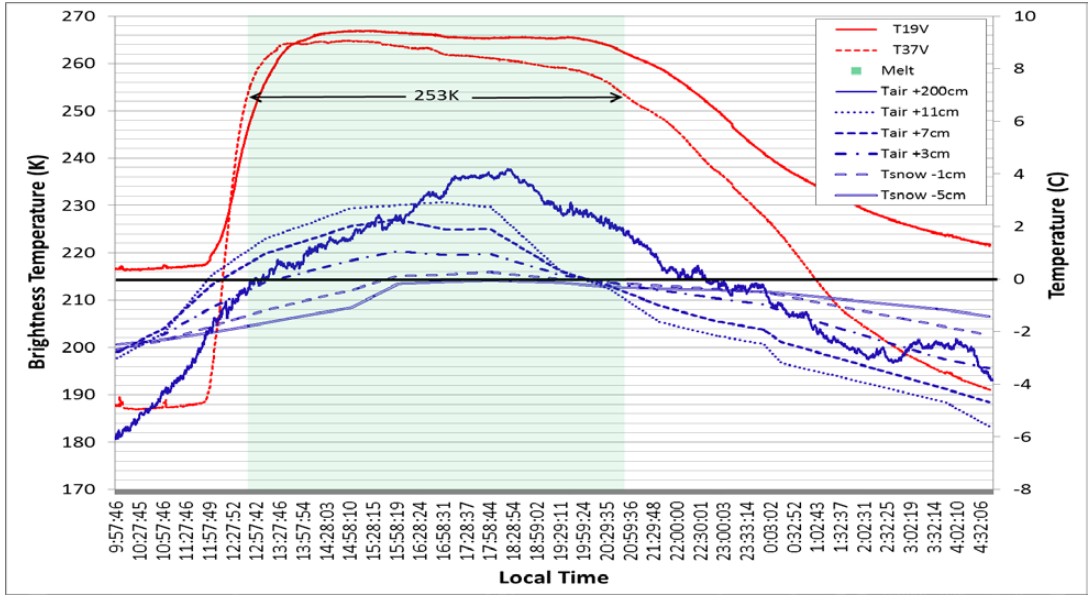

**Figure 4**. Time series of the surface-based radiometer $T_B$ and the air/snow temperature measurements recorded during the April 12-13, 2010 diurnal melt event. The green shaded region highlights the period when the winter $T_BD$ algorithm successfully detected a winter melt event, the onset of which coincides very closely with the 2 m air temperature sensor.




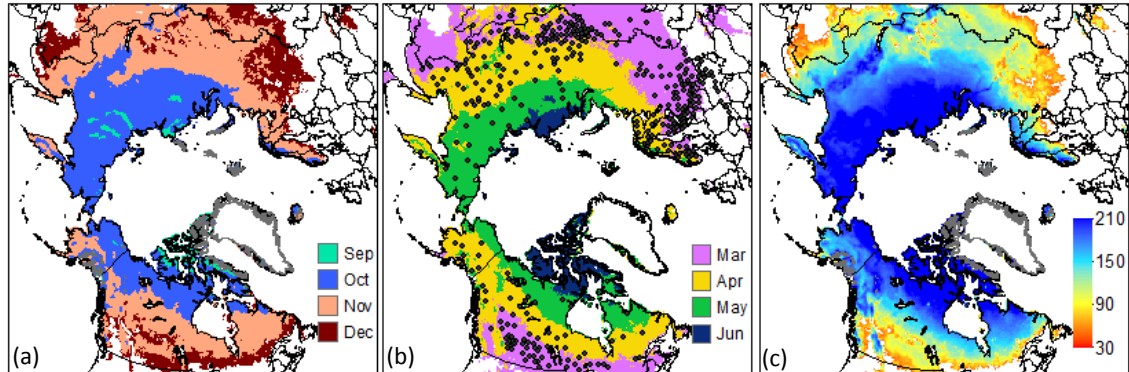

**Figure 5**. The mean main snow onset date in fall (a), main melt onset date in spring (b), and mean winter period duration (days) (c) during the period 1988-2013. The black dots in (b) represent WMO weather stations used for algorithm development and evaluation.



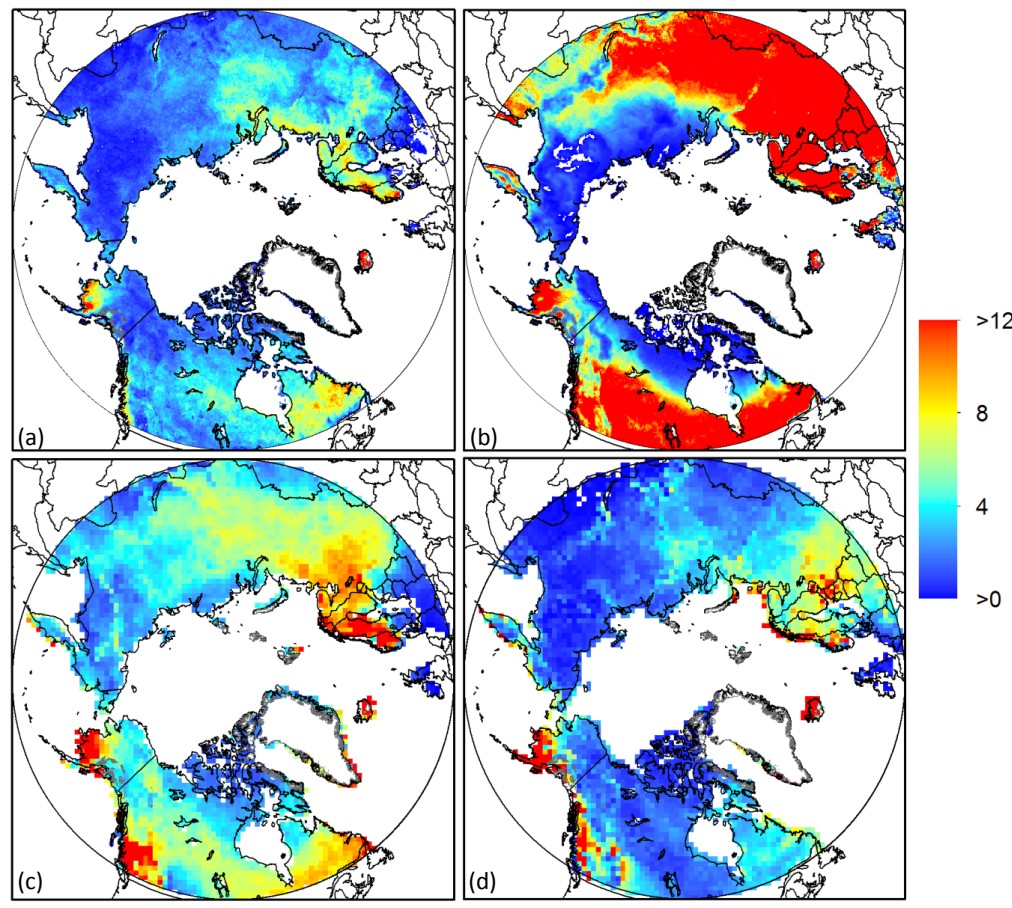

**Figure 6**. The average annual number of melt days over 1988-2013 from (a) PMW using a varying winter period; (b) PMW using a fixed winter period (November to April); (c) ERA-Interim; and (d) MERRA.

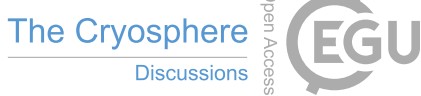

**Figure 7**. Monthly mean number of melting days from PMW during the period 1988-2013.





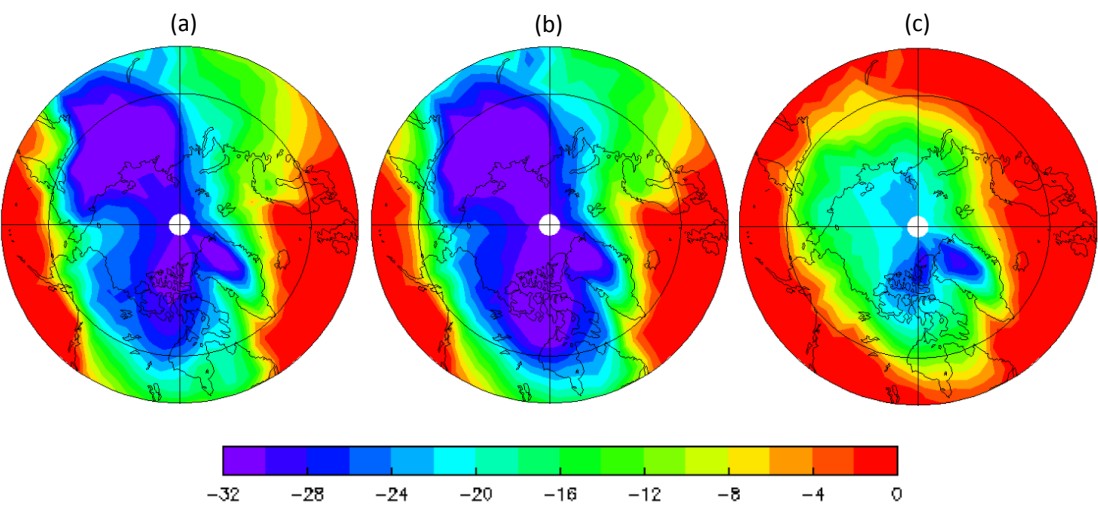

**Figure 8**. The climatological mean surface air temperature from CRUTem4 during the period 1961-1990 for (a) December, (b) February, and (c) April.



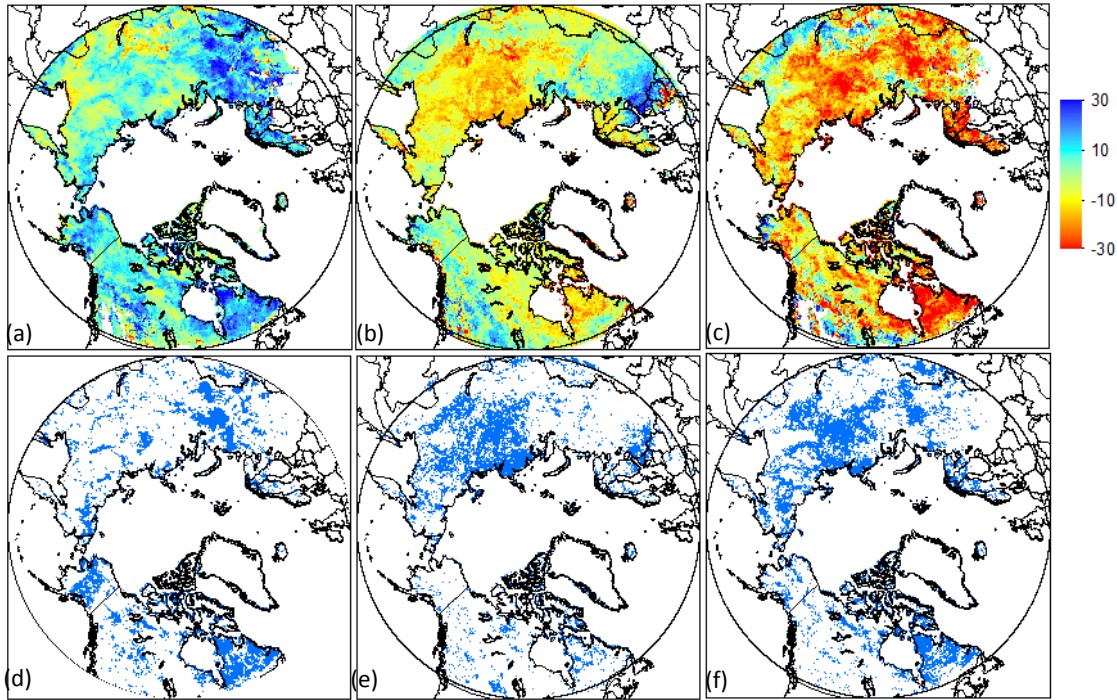

**Figure 9**. Mann-Kendall trends (days/26yr) over the period 1988-2013 in (a) MSOD, (b) MMOD, (c) WPD. Grid cells with trends statistically significant at the 90% level are shown in (d) MSOD, (e) MMOD, and (f) WPD.





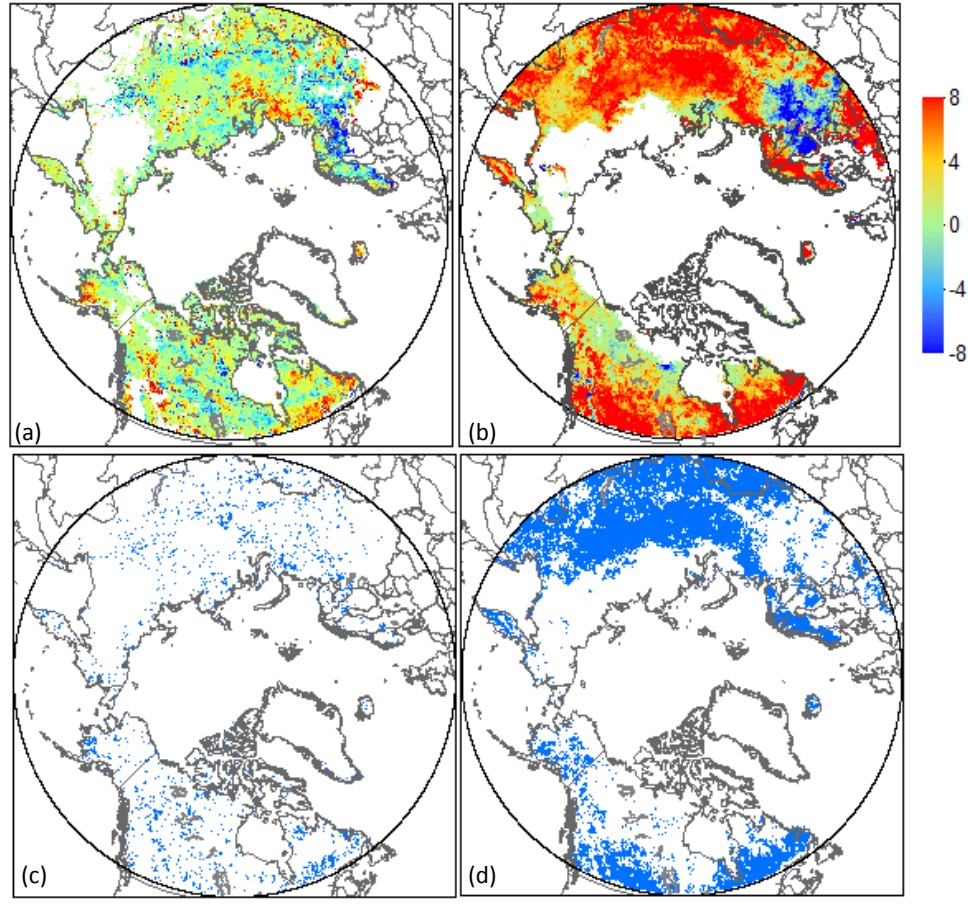

**Figure 10**. Mann-Kendall trends (days/26yr) over the period 1988-2013 in the number of winter melt days from (a) PMW; (b) PMW-fixed; (c) and (d) show grid cells with trends statistically significant at the 90% level in (a) and (b) respectively.




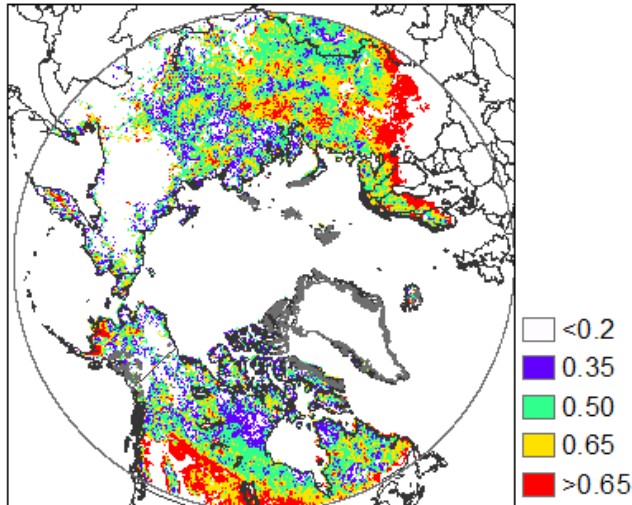

**Figure 11**. The correlation coefficient between number of melt days and the duration of winter period from PMW during 1988-2013.





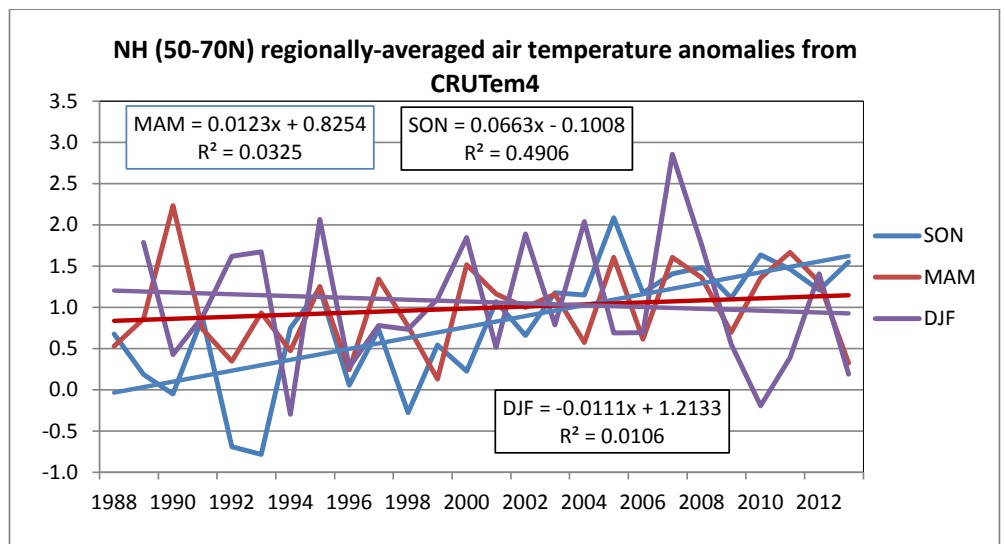

**Figure 12**. Time series of surface air temperature and trends in Northern Hemisphere land areas during the period 1988-2013. Note that the September-October-November (SON) period warmed more than the March-April-May (MAM) and December-January-February (DJF) periods.



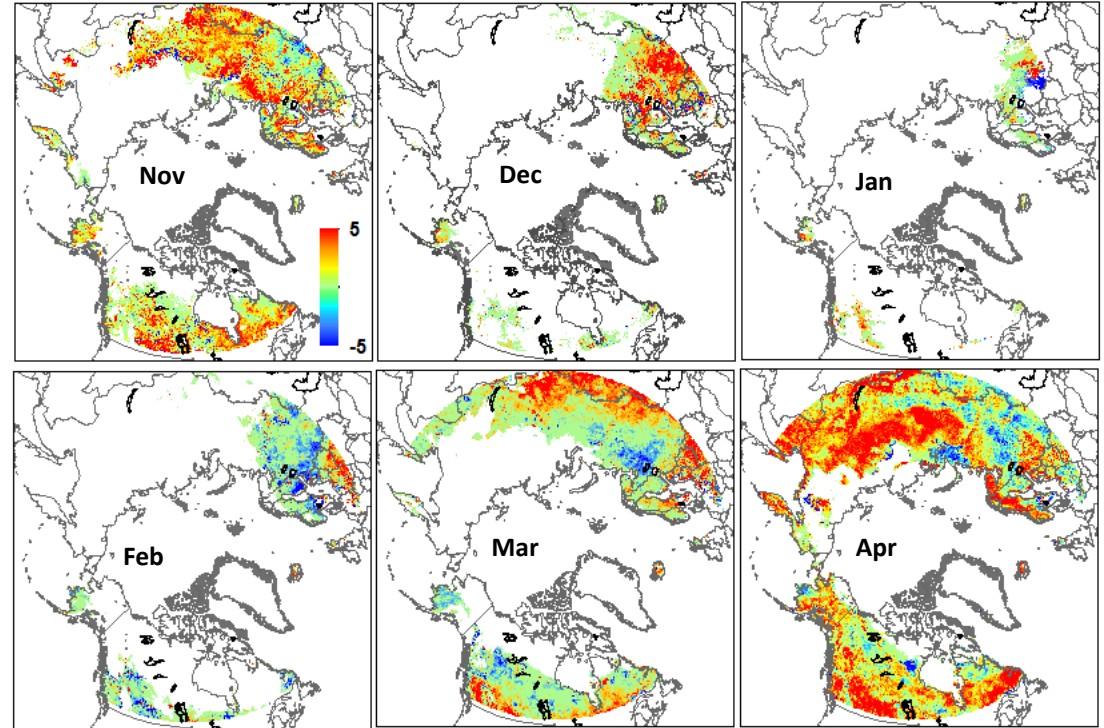

**Figure 13.** Mann-Kendall trends (days/26yr) in the number of melt days derived by PMW-fixed from November to April during the period 1988-2013.