# Peer review of "Frequency and distribution of winter melt events from passive microwave satellite data in the pan-Arctic, 1988-2013"

_The Cryosphere, 2016_

## Referee Comment (RC1) · Anonymous Referee #1 · 8 Jul 2016

Passive microwave satellite data are frequently used to identify changes of snow properties, especially timing of melt. Mostly spring snowmelt timing is addressed in non-glaciated areas and melt days are extracted over glaciers and ice sheets. This study seeks to detect melt days over non-glaciated snow covered areas as well as investigates options for detection of snow cover (winter) start and end. A range of weaknesses of the approach are revealed by comparison to in situ measurements. An interpretation of trends and patterns are provided but usefulness questionable (see comment below). Mid-winter patterns have been described before, as well as snow duration analyses. Kim et al. 2011 have also used SSMI to detect surface status.

It is stated in the introduction that little is known about the spatial and temporal vari-

ability of winter melt events at Pan-Arctic scale (line 44). There are however a number of re-analyses studies available on this topic (e.g. Liston and Hiemstra 2011, Rennert 2009) and also from active microwave satellite data (Bartsch 2010). The observed patterns found in the presented study agree with the above studies, what is not addressed in the discussion.

There are inconsistencies regarding terminology. The title and abstract refer to 'events', the text/method to melt days. Events might be of several days of duration. In addition, only afternoon data are used. The paper thus presents an account of melt 'afternoons'. The title and abstract should be revised and adjusted to reflect this.

The usefulness of the trend analyses of late afternoon melts is questionable. The authors should also include the morning measurements in order to increase the detection capability. Mid-winter melt events are not bound to diurnal-variations. This would still miss out events, but increase the number of samples. Previous studies have actually chosen the characteristic refreeze-pattern instead of melt detection (e.g. Bartsch et al. 2010). Detection of refreeze allows the inclusion of very short melt events which cannot be detected themselves due to the satellite data sampling intervals.

The abstract includes the information that results are compared to in situ measurements, but not the outcome. Especially short events from ROS are not detected, which are of major interest for wild live and climate change studies. The failure in such cases demonstrates the shortcoming of the approach to use melt only.

How does the performance compare to melt day detection performance commonly used over ice sheets and glaciers?

How does the approach of melt detection compare to results from Kim et al. 2011 ( SSMI) or Naeimi et al. 2012 (ASCAT)? Kim et al 2011 showed that a dynamic threshold is needed.

Kim et al. 2012 also analyze passive microwave trend analyses for snow cover. How
do patterns compare?

Other comments

Line 48: Semmens et al. 2013 also demonstrated the importance of fog

Line 60: add e.g. before the list of references as there are many more studies published on this Topic

Line 63: Semmens et al. 2013 also used passive microwave data. Grennfell and Putkonen also used passive microwave data

Section 3.2. – results agree with Bartsch 2010

Additional references

Kim Y, Kimball J S, Zhang K and McDonald K C 2012 Satellite detection of increasing northern hemisphere non-frozen seasons from 1979 to 2008: implications for regional vegetation growth Remote Sens. Environ.121472–87

Bartsch, A. (2010): Ten Years of SeaWinds on QuikSCAT for Snow Applications. Remote Sens. 2010, 2(4), 1142-1156; doi:10.3390/rs2041142;

Naeimi, V., Paulik, C., Bartsch, A., Wagner, W., Kidd, R., Boike, J. and K. Elger (2012): ASCAT Surface State Flag (SSF): ASCAT Surface State Flag (SSF): Extracting Information on Surface Freeze/Thaw Conditions From Backscatter Data Using an Empirical Threshold-Analysis Algorithm. IEEE Transactions on Geoscience and Remote Sensing. DOI: 10.1109/TGRS.2011.2177667.

---

## Referee Comment (RC2) · Anonymous Referee #2 · 28 Jul 2016

The manuscript describes a climatology of snow melt days across the Arctic or land regions poleward of 50N using passive microwave observations. They also validate their results against reanalysis datasets and from station data/snowpit surveys. They find that snowmelt days are relatively rare (a week or less) over the winter period. They do find that snowmelt days are positively correlated with length of the winter season (defined as the period of a stable snowpack) and that there are only weak trends in snowmelt days.

This is a strong team of topic experts, a well-written manuscript and the analysis was expertly executed. The topic is of interest and the manuscript a worthy contribution to the cryosphere community and has relevance to climate change as well. I have very

few comments to add to improve the manuscript. My few minor comments are listed below.

I did see that another reviewer found inconsistencies in the definition of melt events. I was not bothered by potential inconsistencies though it is probably best for the authors to clarify their definitions.

I recommended that the manuscript be accepted pending minor revisions.

Minor comments: 1. Line 110 – the authors state that they filled data gaps through linear interpolation form adjacent days. However the authors mentioned above the technique for detecting water is robust because there are large variations in TB depending on the presence of water. Therefore simply linear interpolating would be problematic near dates of snowmelt?

2. Figure 5 – in panels 5a and 5b why not show MSOD and NMOD as day of year rather than as month?

3. Is it possible that the reanalysis products (especially ERA-Interim) in general have more snowmelt days because they are sampled four times daily and the PMW only once a day? This should be checked.

4. Figure 8 – why use a temperature climatology of 1961-1990 which is colder than the period of the passive microwave data set of 1988-2013? Preferably an overlapping period should be used for the temperature climatology or even 1981-2010.

5. Figure 12 – the results presented in the figure where temperatures are warming in the fall and spring but not winter across the Northern Hemisphere landmasses is not a new result but is very similar to seasonal temperature trends shown in Cohen et al. 2012.

Reference: Cohen, J., J. Furtado, M. Barlow, V. Alexeev and J. Cherry 2012: Asymmetric seasonal temperature trends. Geophys. Res. Lett., 39, L04705, doi:10.1029/2011GL050582.

---

## Referee Comment (RC3) · Anonymous Referee #3 · 16 Aug 2016

Review of "Frequency and distribution of winter melt events from passive microwave satellite data in the pan-Arctic, 1988-2013"

By L. Wang, P. Toose, R. Brown, and C. Derksen

Submitted to *The Cryosphere*

Manuscript Number: tc-2016-126

Summary: In this paper, the authors undertake an analysis of mid-winter snow melt events across land areas of the pan-Arctic domain above 50°N using microwave remote sensing. An algorithm is developed to infer liquid water in snowpacks using variations in surface brightness temperatures from SSM/I and SSMIS over 1988-2013. Mid-winter melt events are relatively rare with ≤7 occurrences (days) each year across most areas under study, with higher frequencies in temperate regions. The spatial patterns in winter snow melt events inferred from air temperature obtained from reanalysis products concur with those detected by the microwave remote sensing data. Further analyses reveal few statistically significant trends in winter melt events with the notable exception of northern Europe.

This is an interesting paper with novel results and it should be suitable for publication in *The Cryosphere* following some moderate revisions. My report provides guidance on how the paper should be revised prior to publication:

General Comments:

1) In-text references do not follow the format used by *The Cryosphere*, i.e. round rather than square brackets should be used for references.
2) Has validation of the proposed algorithm been performed in regions other than Canada and Finland, such as Russia and Alaska?
3) At times snow melt events occur just below the surface of the snowpack − is the proposed methodology able to detect such events?
4) The results presented in this paper focus on terrestrial snowpacks − can the methodology also be applied to snow on sea ice?
5) How reliable is the algorithm when applied to complex terrain such as the western Cordillera of North America?
6) If only the afternoon overpasses are used to infer snow melt events across the pan-Arctic, how are melt events during other times of the day accounted for?
7) Probability values should be reported for all correlation coefficients presented in the paper.
8) The findings of recent rising air temperatures during fall (SON) with no trends in winter (DJF) and spring (MAM) across the Northern Hemisphere seem to contradict results from other studies (see Figure 12). These results should be placed into context (time period and area of interest). Why are temperature trends not reported only for the domain of study (i.e. pan-Arctic land areas above 50°N) for comparison with the snow melt analyses? Why are the seasonal air

temperature trends not inferred from the Mann-Kendall test instead of linear regressions? Probability values for these trends should also be reported.

9) Further to this, how reliable are trend analyses for a rather short (25 years, 1988-2013) period of study? Are the reported trends greater than the variability experienced over the period of study, i.e. is the signal greater than the noise in the data?

10) The authors should consider suggestions for future work in the final paragraph of Section 4.

Specific Comments:

1) P. 1, line 12: Insert "GHz" after "19".
2) P. 1, line 19: Replace "7" with "seven".
3) P. 1, line 22: "ERA" and "MERRA" are not defined.
4) P. 2, line 34: Insert a comma after "events".
5) P. 5, line 104: Define "EASE".
6) P. 6, line 126: Insert "GHz" after "19" and insert a space in the second "37 GHz".
7) P. 7, line 151: Add a comma after "e.g."
8) P. 8, line 170: Change to "one week".
9) P. 8, line 195: Insert a comma after "disappearance".
10) P. 9, lines 197/198: Delete "degree" and define acronyms used here.
11) P. 9, line 203: Why are 30-day moving averages of daily mean air temperatures used here for analysis?
12) P. 10, line 224: Insert a space in "Table 2".
13) P. 11, line 246: Delete the space in "0°C".
14) P. 11, line 248: Should this be "1 cm" instead of "-1 cm"? Replace the contraction "didn't" with "did not" and delete the space in "0°C".
15) P. 11, line 250: Delete the space in "0°C".
16) P. 12, lines 269/270: More information in the Methods must be provided on the selection of Daring Lake and La Grande IV as areas to test the algorithm to detect snow melt events. Provide for instance the province/territory where these locations are found and a brief description of their environment (vegetation, physiography, etc.) What does "La Grande IV" mean?
17) P. 13, line 285: Revise to "(Figure 5c). A pixel-wise"…
18) P. 13, line 286: Delete the second "winter".
19) P. 14, line 301: Insert a comma after "e.g.".
20) P. 14, lines 316 to 318: Are any of these trends statistically-significant? It is difficult to interpret linear trends when associated probability values are not provided. Figure captions for trend analyses do report a statistical significance of 90% and as such the Methods section must discuss use of this level as definition of statistically-significant trends.
21) P. 14, line 319: Delete "are shown in" and insert brackets in "(Figure 9)."
22) P. 15, line 321: Avoid tentative language such as "tends".
23) P. 15, line 323: Delete "period".
24) P. 15, line 327: Again avoid the use of tentative language.

25) P. 15, line 334: What is the probability value for the correlation coefficient reported here?
26) P. 15, line 336: Replace "are" with "is".
27) P. 16, line 348: Revise to "lasts".
28) P. 16, line 363: Change to "northern".
29) P. 17, line 370: Replace "which" with "that".
30) P. 17, line 383: Replace "which tend to" by "that produce".
31) P. 17, line 386: Delete "which revealed".
32) P. 18, line 404: Should this be "pan-Arctic"?
33) P. 18, line 405: Any thoughts on possible future work that could be added here?
34) P. 18, line 409: Replace "which" by "that".
35) P. 28, Table 1: How does the change in SSM/I orbital overpass from descending (July 1988 to December 1991) to ascending affect the results presented in this study?
36) P. 31, Figure 2: Are snow pit data available for this site in Finland, as presented in Figure 3 for Manitoba?
37) P. 32, Figure 3: If possible, this figure should have the same format (two panels) as shown in Figure 2 for consistency between them. Are Tmin and Tmax not available for this site?
38) P. 33, Figure 4: The caption should specify the location where these time series results apply.
39) P. 34, Figure 5: How do these results compare to those presented by Choi et al. (2010)?
40) P. 35, Figure 6: The color scale should be identified as "Days".
41) P. 36, Figure 7: Why are results for June not presented here? Please define the color scale here as well.
42) P. 37, Figure 8: What are the units for the color scale? Why are these results presented and how relevant are they to those on the detection of snow melt events from microwave remote sensing?
43) PP. 38/39, Figures 9 and 10: The text must specify what level of significance trends are reported at. Insert "Days" for the color scales here too.
44) P. 40, Figure 11: What are the probability values for the correlation coefficients presented here?
45) P. 41, Figure 12: This figure could be improved by using a program other than Excel for plotting. The y-axis lacks a title and units.

References:

Choi, G., Robinson, D. A., and Kang, S.: Changing Northern Hemisphere snow seasons, J. Climate, 23, 5305-5310, 2010.

---

## Author Comment (AC1) · 13 Sep 2016

**Response to reviewer comments**

We thank all reviewers for their helpful comments. Please find below our responses in blue.

**Response to Reviewer #1**

Passive microwave satellite data are frequently used to identify changes of snow properties, especially timing of melt. Mostly spring snowmelt timing is addressed in non- glaciated areas and melt days are extracted over glaciers and ice sheets. This study seeks to detect melt days over non-glaciated snow covered areas as well as investigates options for detection of snow cover (winter) start and end. A range of weak- nesses of the approach are revealed by comparison to in situ measurements. An interpretation of trends and patterns are provided but usefulness questionable (see comment below). Mid-winter patterns have been described before, as well as snow duration analyses. Kim et al. 2011 have also used SSMI to detect surface status. It is stated in the introduction that little is known about the spatial and temporal variability of winter melt events at Pan-Arctic scale (line 44). There are however a number of re-analyses studies available on this topic (e.g. Liston and Hiemstra 2011, Rennert 2009) and also from active microwave satellite data (Bartsch 2010). The observed patterns found in the presented study agree with the above studies, what is not addressed in the discussion.

We thank the reviewer for the comments, but what the reviewer interprets as weaknesses in our methodology, we see as inherent limitations of the PMW sensor that are clearly noted and discussed in the paper.

We have removed Line44, and added Bartsch 2010 in the introduction and discussion. The other references are already cited in the paper. The Kim et al [2011] study was carried out for landscape Freeze/Thaw (FT) detection and they did not differentiate the FT signal coming from snow-covered versus snow-free surfaces. Their results are therefore not comparable with this study, which focuses only on snow-covered regions for winter-snowmelt detection.

There are inconsistencies regarding terminology. The title and abstract refer to 'events', the text/method to melt days. Events might be of several days of duration. In addition, only afternoon data are used. The paper thus presents an account of melt 'afternoons'. The title and abstract should be revised and adjusted to reflect this.

The algorithm does detect winter melt events, but we summarized the results as the number of melt days to avoid the issue of event splitting that can occur with the algorithm. We have now explicitly explained this strategy in Lines 190-192.

The usefulness of the trend analyses of late afternoon melts is questionable. The authors should also include the morning measurements in order to increase the detection capability. Mid-winter melt events are not bound to diurnal-variations. This would still miss out events, but increase the number of samples. Previous studies have actually chosen the characteristic refreeze-pattern instead of melt detection (e.g. Bartsch et al. 2010). Detection of refreeze allows the inclusion of very short melt events which cannot be detected themselves due to the satellite data sampling intervals.

Good point. We have included melt detection from the morning orbits and updated all the results. This has indeed increased the number of melt days in some temperate climate regions (e.g., southern Alaska and northern Europe). However, it has not resulted in much change in either the spatial distribution patterns or the trend analyses.

The abstract includes the information that results are compared to in situ measurements, but not the outcome. Especially short events from ROS are not detected, which are of major interest for wild live and climate change studies. The failure in such cases demonstrates the shortcoming of the approach to use melt only.

We have modified the abstract to include the validation results.

Bartsch et al. 2010 used the increase of backscatter to detect refreeze events from QuikSCAT. However, the record of QuikSCAT is too short for trend analyses. The increase in the spectral gradient of 19 and 37 GHz from the SSM/I data ($T_BD$) has been widely used for snow water equivalent retrievals [e.g., Chang et al., 1987], which is also used to determine the main snow onset date in the fall in this study. Although all the melt/refreeze events during the winter are associated with a decrease followed by an increase in $T_BD$ (Fig. 2), not all increases in $T_BD$ can be attributed to refreeze events (some are due to snow accumulation). Similar ambiguities apply for refreeze events detection from QuikSCAT data [Bartsch et al., 2010].

This study focuses on winter melt detection, which occurs more often than ROS [Bartsch et al., 2010; Cohen et al., 2015]. With regard to ROS, we have re-examined all events included in Table 2, and added the following to Section 3.1 (Lines 296-301):

"Out of all twelve melt events investigated, six events coincided with observed ROS. Of the six ROS events, half were associated with successful satellite detection. Those ROS events that were successfully detected were followed by a continued warming of air temperatures that likely delayed the re-freezing of the liquid water in the snow. Those ROS events that were not detected fall under the category of a short duration melt event and thus are not detectable, as described above."

How does the performance compare to melt day detection performance commonly used over ice sheets and glaciers?

Melt over ice sheets and glaciers usually occur during the spring/summer melt season (e.g., Tedesco, 2007) which is the time of year we exclude for detecting winter melt events. Thus it is not appropriate to compare the performance of winter melt detection over seasonal snow to those on ice sheets and glaciers. See also Lines 202-204.

How does the approach of melt detection compare to results from Kim et al. 2011 (SSMI) or Naeimi et al. 2012 (ASCAT)? Kim et al 2011 showed that a dynamic threshold is needed.

Kim et al [2011] used a seasonal threshold approach and optimized the threshold values using reanalysis air temperatures. In this sense, the remote sensing retrievals are 'calibrated' using air temperature information. As mentioned earlier, Kim et al [2011] carried out landscape FT detection at a global scale, and did not differentiate the FT signal from snow-covered vs snow-free surfaces. Naeimi et al. 2012 (ASCAT) only showed surface state flags of frozen/unfrozen or snowmelt, they did not show the number of melt days over the winter. Thus the results from the two studies are not comparable with winter melt day results in the current study. Our method also uses dynamic pixel-dependent thresholds to determine the main snow onset, the main melt onset, and the winter melt days. We have clarified this in Section 2.2.

Kim et al. 2012 also analyze passive microwave trend analyses for snow cover. How do patterns compare?

Kim et al [2012] used a similar approach as in Kim et al [2011] and thus did not differentiate the FT signal from snow-covered vs snow-free surfaces. Furthermore, Kim et al [2012] only showed

trends for the non-frozen period (as indicated in the title), which is not comparable with the winter melt day trends from this study.

Other comments

Line 48: Semmens et al. 2013 also demonstrated the importance of fog

A reference to fog by Semmens et al [2013] is included in the revised manuscript in line 52.

Line 60: add e.g. before the list of references as there are many more studies published on this Topic
Done

Line 63: Semmens et al. 2013 also used passive microwave data. Grennfell and Putkonen also used passive microwave data

We have modified the sentence and included Grennfell and Putkonen, 2008.

Section 3.2. – results agree with Bartsch 2010

We have added this in the discussion Section.

Additional references

Kim Y, Kimball J S, Zhang K and McDonald K C 2012 Satellite detection of increasing northern hemisphere non-frozen seasons from 1979 to 2008: implications for regional vegetation growth Remote Sens. Environ.121472–87

Bartsch, A. (2010): Ten Years of SeaWinds on QuikSCAT for Snow Applications. Remote Sens. 2010, 2(4), 1142-1156; doi:10.3390/rs2041142;

Naeimi, V., Paulik, C., Bartsch, A., Wagner, W., Kidd, R., Boike, J. and K. Elger (2012): ASCAT Surface State Flag (SSF): ASCAT Surface State Flag (SSF): Extracting Infor- mation on Surface Freeze/Thaw Conditions From Backscatter Data Using an Empirical Threshold-Analysis Algorithm. IEEE Transactions on Geoscience and Remote Sens- ing. DOI: 10.1109/TGRS.2011.2177667.

---

## Author Comment (AC2) · 13 Sep 2016

**Response to reviewer comments**

We thank all reviewers for their helpful comments. Please find below our responses in blue.

**Response to reviewer #2**

The manuscript describes a climatology of snow melt days across the Arctic or land regions poleward of 50N using passive microwave observations. They also validate their results against reanalysis datasets and from station data/snowpit surveys. They find that snowmelt days are relatively rare (a week or less) over the winter period. They do find that snowmelt days are positively correlated with length of the winter season (defined as the period of a stable snowpack) and that there are only weak trends in snowmelt days.

This is a strong team of topic experts, a well-written manuscript and the analysis was expertly executed. The topic is of interest and the manuscript a worthy contribution to the cryosphere community and has relevance to climate change as well. I have very few comments to add to improve the manuscript. My few minor comments are listed below.

I did see that another reviewer found inconsistencies in the definition of melt events. I was not bothered by potential inconsistencies though it is probably best for the authors to clarify their definitions.

We thank the reviewer for the positive comments. We have added some additional explanation in Lines 190-192 to clarify the melt event/day issue.

I recommended that the manuscript be accepted pending minor revisions.

Minor comments: 1. Line 110 – the authors state that they filled data gaps through linear interpolation form adjacent days. However the authors mentioned above the technique for detecting water is robust because there are large variations in TB depending on the presence of water. Therefore simply linear interpolating would be problematic near dates of snowmelt?

Good point. Filling data gaps through linear interpolation from adjacent days will certainly bring some uncertainties to the detection results. However, this should have been somewhat mitigated by using both $T_BD$ and $T_B37V$ for melt detection (see section 2.2 Lines 156-158). In addition, the large differences of $T_BD$ and $T_B37V$ for days with melt and freeze conditions (Fig. 2) would limit false detection for days filled by linear interpolation.

The Kim et al. [2011] study was for freeze/thaw detection from the SSM/I data globally (thus they had more data gaps than this study). They also used linear interpolation from adjacent days for gap filling as in this study.

2. Figure 5 – in panels 5a and 5b why not show MSOD and NMOD as day of year rather than as month?

We show MSOD and MMOD as month in Fig. 5 so that it is easier to understand the spatial distribution patterns of monthly mean number of melt days described in Section 3.2 and shown in Fig. 7. In addition we describe the spatial distribution of MSOD and MMOD by months in Section 3.2.

3. Is it possible that the reanalysis products (especially ERA-Interim) in general have more snowmelt days because they are sampled four times daily and the PMW only once a day? This should be checked.

The reanalysis-based method that we employed, used the daily mean temperature to estimate melt events so the potential impact of the more frequent sub-daily sampling is dampened. We also now use both morning and afternoon overpass to detect winter melt from the satellite data, making the satellite results more comparable to those of the daily reanalysis data. Using both the morning and afternoon satellite passes results in some increase in melt days from the satellite mainly in temperate climate regions, such as southern Alaska and northern Europe (Fig. 6), however, the increases are too small to fully resolve the different melt days from the satellite and reanalysis (especially ERA-I).

4. Figure 8 – why use a temperature climatology of 1961-1990 which is colder than the period of the passive microwave data set of 1988-2013? Preferably an overlapping period should be used for the temperature climatology or even 1981-2010.

This figure was removed from the paper as it was not considered essential and the climatology can be readily generated from existing gridded observational or reanalysis datasets.

5. Figure 12 – the results presented in the figure where temperatures are warming in the fall and spring but not winter across the Northern Hemisphere landmasses is not a new result but is very similar to seasonal temperature trends shown in Cohen et al. 2012.

Reference: Cohen, J., J. Furtado, M. Barlow, V. Alexeev and J. Cherry 2012: Asymmetric seasonal temperature trends. Geophys. Res. Lett., 39, L04705, doi:10.1029/2011GL050582.

Thank you for noting. We have cited the reference in the paper.

---

## Author Comment (AC3) · 13 Sep 2016

**Response to reviewer comments**

We thank all reviewers for their helpful comments. Please find below our responses in blue.

**Response to Reviewer #3**

Summary:    In this paper, the authors undertake an analysis of mid-winter snow melt events across land areas of the pan-Arctic domain above 50°N using microwave remote sensing. An algorithm is developed to infer liquid water in snowpacks using variations in surface brightness temperatures from SSM/I and SSMIS over 1988-2013. Mid-winter melt events are relatively rare with ≤7 occurrences (days) each year across most areas under study, with higher frequencies in temperate regions. The spatial patterns in winter snow melt events inferred from air temperature obtained from reanalysis products concur with those detected by the microwave remote sensing data. Further analyses reveal few statistically significant trends in winter melt events with the notable exception of northern Europe.

This is an interesting paper with novel results and it should be suitable for publication in *The Cryosphere* following some moderate revisions. My report provides guidance on how the paper should be revised prior to publication:

We thank the reviewer for the positive feedback.

General Comments:

1)  In-text references do not follow the format used by *The Cryosphere*, i.e. round rather than square brackets should be used for references.

Square brackets are allowed according to instructions on TC website: http://www.the-cryosphere.net/for_authors/manuscript_preparation.html

2)  Has validation of the proposed algorithm been performed in regions other than Canada and Finland, such as Russia and Alaska?

Yes – The algorithm was developed/validated with observations at the WMO weather stations across the pan-Arctic as shown in Figure 5b. Note the validation results using the weather station data are presented in the Data and Method Section (Lines 160-169). However, the in situ field measurements (snow survey and surface-based radiometer data) were only collected by the authors in Canada.

3)  At times snow melt events occur just below the surface of the snowpack – is the proposed methodology able to detect such events?

This is probably not common during the winter. The melt detection algorithm is based on the sensitivity of microwave signal to the appearance of liquid water in the snowpack (surface or subsurface) - there is a sharp decrease in $T_BD$ from dry to wet snow transition. Thus it should be able to detect subsurface melt events as well. However, detection of sub-surface melting is similar to a mixed-pixel effect (presumably dry/frozen surface and wet melted sub-surface), and thus would be hard to quantify at the satellite scale. Figure 4 provides some evidence that the F/T signal from uneven surface and sub-surface re-freeze likely becomes muted relative to the

initial onset of melt. See the Results section on lines 281-283. We have also added the following sentence in the Discussion and Conclusions Section (Lines 378-379).

"The algorithm should also be able to detect subsurface melt events although this aspect was not evaluated in this paper."

4) The results presented in this paper focus on terrestrial snowpacks – can the methodology also be applied to snow on sea ice?

Good question. Similar channel difference approaches have also been used for snowmelt onset detection over the Arctic sea ice [e.g., Drobot and Anderson, 2001]. However, the emissivities of first-year sea ice are different than that of multiyear sea ice, and the emissivities over multiyear sea ice can have a large range due to the varied histories of the ice floes. These complicate the detection of snowmelt over sea ice, so we do not recommend the use of the algorithm developed in this study for melt detection over sea ice. A multiple indicators approach was developed in Markus et al [2009] for melt/refreeze detection over the Arctic sea ice. We have added the above to the Discussion and Conclusion section (Lines 379-386).

Drobot, S. D., and Anderson, M. R.: An improved method for determining snowmelt onset dates over Arctic sea ice using scanning multichannel microwave radiometer and Special Sensor Microwave/ Imager data, J. Geophys. Res., 106, 24,033 – 24,049, doi:10.1029/2000JD000171, 2001.

5) How reliable is the algorithm when applied to complex terrain such as the western Cordillera of North America?

Good point. The algorithm is based on the large difference of $T_BD$ for dry snow versus wet snow (~30K), however, the range of $T_BD$ can be much smaller (~10K) in areas with deep snow and complex terrain [Tong et al., 2010]. In-addition, changes in elevation and terrain aspect can have profound influence on air temperatures at the local scale, resulting in dramatic temperature differences over very short distances. Therefore the use of coarse resolution passive microwave satellites to detect melt events in complex terrain is not recommended. The performance of the algorithm in these areas may have a relatively large uncertainty that needs to be further evaluated. This can be an area of future work. We have added this in Section 4.

6) If only the afternoon overpasses are used to infer snow melt events across the pan- Arctic, how are melt events during other times of the day accounted for?

Good point. We have now included snow melt events from the morning overpasses as well.

7) Probability values should be reported for all correlation coefficients presented in the paper. Done

8) The findings of recent rising air temperatures during fall (SON) with no trends in winter (DJF) and spring (MAM) across the Northern Hemisphere seem to contradict  results  from  other studies  (see  Figure  12).  These  results  should  be placed into context (time period and area of interest). Why are temperature trends not reported only for the domain of study (i.e. pan-Arctic land areas above 50°N) for comparison with  the  snow melt  analyses? Why  are  the seasonal  air temperature trends  not  inferred  from  the Mann-Kendall  test  instead  of  linear regressions? Probability values for these trends should also be reported.

To be consistent, we have computed the seasonal air temperature trends using the Mann-Kendall test from CRUTem4 data and included the results in the text. The results are very similar

to those from linear regressions. We have provided a trend map for the winter season (Figure 11).

9) Further to this, how reliable are trend analyses for a rather short (25 years, 1988-2013) period of study? Are the reported trends greater than the variability experienced over the period of study, i.e. is the signal greater than the noise in the data?

Good point. We now explicitly acknowledge this in Lines 405-408. The question of signal/noise is taken account of in the test for trend statistical significance.

10) The authors should consider suggestions for future work in the final paragraph of Section 4.

We have added a couple of sentences at the end of the final paragraph for future work.

Specific Comments:

1) P. 1, line 12: Insert "GHz" after "19".
Done

2) P. 1, line 19: Replace "7" with "seven".
We have replaced 7 with one week

3) P. 1, line 22: "ERA" and "MERRA" are not defined.
These are very common names, for briefness we do not define them in the abstract.

4) P. 2, line 34: Insert a comma after "events".
Done

5) P. 5, line 104: Define "EASE".
Done

6) P. 6, line 126: Insert "GHz" after "19" and insert a space in the second "37 GHz".
Done

7) P. 7, line 151: Add a comma after "e.g."
Done

8) P. 8, line 170: Change to "one week".
Done

9) P. 8, line 195: Insert a comma after "disappearance".
Done

10) P. 9, lines 197/198: Delete "degree" and define acronyms used here.
Done

11) P. 9, line 203: Why are 30-day moving averages of daily mean air temperatures used here for analysis?

This is to define the start and end of winter period similar as in the satellite approach. We have modified the sentence to clarify this point.

12) P. 10, line 224: Insert a space in "Table 2".

Done

13) P. 11, line 246: Delete the space in "0°C".
The space is required by the journal.

14) P. 11, line 248: Should this be "1 cm" instead of "-1 cm"? Replace the contraction "didn't" with "did not" and delete the space in "0°C".

The snow temperature is for 1 cm below the surface, so it is -1 cm. We have replaced "didn't" with "did not".

15) P. 11, line 250: Delete the space in "0°C".
See above.

16) P. 12, lines 269/270: More information in the Methods must be provided on the selection of Daring Lake and La Grande IV as areas to test the algorithm to detect snow melt events. Provide for instance the province/territory where these locations are found and a brief description of their environment (vegetation, physiography, etc.) What does "La Grande IV" mean?

The specific locations/provinces of the field sites are provided in Table 2. As indicated in Table 2, the Survey Sites are named after the closest weather station while the actual survey locations are provided in lat/lon. On Line 237 it is noted that the sites are a mix of boreal forest and tundra environments. We chose these locations because of the availability of snowpit survey data with melt/ice crusts recorded in field notes.

17) P. 13, line 285: Revise to "(Figure 5c). A pixel-wise"…
Done

18) P. 13, line 286: Delete the second "winter".
Done

19) P. 14, line 301: Insert a comma after "e.g.".
Done

20) P. 14, lines 316 to 318: Are any of these trends statistically-significant? It is difficult to interpret linear trends when associated probability values are not provided. Figure captions for trend analyses do report a statistical significance of 90% and as such the Methods section must discuss use of this level as definition of statistically-significant trends.
We have added a sentence in the Methods Section to indicate the use of 90% level as definition of statistically-significant trends.

21) P. 14, line 319: Delete "are shown in" and insert brackets in "(Figure 9)."
We have modified the sentence to include information about the significance level.

22) P. 15, line 321: Avoid tentative language such as "tends".
We have modified the sentence.

23) P. 15, line 323: Delete "period".
We prefer to keep the "period" because we're referring to the winter period duration defined in this study, which is different than the commonly used winter season (i.e. DJF).

24) P. 15, line 327: Again avoid the use of tentative language.
Done

25) P. 15, line 334: What is the probability value for the correlation coefficient reported here?

$p < 0.001$, we have added this in the text.

26) P. 15, line 336: Replace "are" with "is".
Done

27) P. 16, line 348: Revise to "lasts".
Done

28) P. 16, line 363: Change to "northern".
Done

29) P. 17, line 370: Replace "which" with "that".
Done

30) P. 17, line 383: Replace "which tend to" by "that produce".
Done

31) P. 17, line 386: Delete "which revealed".
We have modified the sentence.

32) P. 18, line 404: Should this be "pan-Arctic"?
We have removed this sentence.

33) P. 18, line 405: Any thoughts on possible future work that could be added here?
We have added a couple of sentences for future work at the end of the paragraph.

34) P. 18, line 409: Replace "which" by "that".
Done

35) P. 28, Table 1: How does the change in SSM/I orbital overpass from descending (July 1988 to December 1991) to ascending affect the results presented in this study?

Note F-08 descending (July 1988 to December 1991) is for afternoon overpass, which is different than other satellites. We have modified Table 1 to include both the morning and afternoon overpass.

36) P. 31, Figure 2: Are snow pit data available for this site in Finland, as presented in Figure 3 for Manitoba?
No, we choose this site for its multiple melt/refreeze events.

37) P. 32, Figure 3: If possible, this figure should have the same format (two panels) as shown in Figure 2 for consistency between them. Are Tmin and Tmax not available for this site?

Note this figure shows hourly air temperature, so it is impossible to make it the same as in Figure 2, which shows daily air temperature.

38) P. 33, Figure 4: The caption should specify the location where these time series results apply.
Done

39) P. 34, Figure 5: How do these results compare to those presented by Choi et al. (2010)?

Choi et al. [2010] only presented time series of the average snow season duration over the Northern Hemisphere during 1972-2007, not the spatial distribution. Since both the study area and the time period are different between Choi et al. [2010] and this study, it is impossible to compare the results.

40) P. 35, Figure 6: The color scale should be identified as "Days".
Done

41) P. 36, Figure 7: Why are results for June not presented here? Please define the color scale here as well.

Good point. Results for June are now included, color scale defined.

42) P. 37, Figure 8: What are the units for the color scale? Why are these results presented and how relevant are they to those on the detection of snow melt events from microwave remote sensing?

This figure was removed from the paper as it was not considered essential and the climatology can be readily generated from existing gridded observational or reanalysis datasets.

43) PP. 38/39, Figures 9 and 10: The text must specify what level of significance trends are reported at. Insert "Days" for the color scales here too.
Done

44) P. 40, Figure 11: What are the probability values for the correlation coefficients presented here?
We have included the significant level in the caption and text.

45) P. 41, Figure 12: This figure could be improved by using a program other than Excel for plotting. The y-axis lacks a title and units.
We have modified this figure (now figure 11).

References:

Choi, G., Robinson, D. A., and Kang, S.: Changing Northern Hemisphere snow seasons, J. Climate, 23, 5305-5310, 2010.